# Fumonisins: Impact on Agriculture, Food, and Human Health and their Management Strategies

**DOI:** 10.3390/toxins11060328

**Published:** 2019-06-07

**Authors:** Madhu Kamle, Dipendra K. Mahato, Sheetal Devi, Kyung Eun Lee, Sang G. Kang, Pradeep Kumar

**Affiliations:** 1Department of Forestry, North Eastern Regional Institute of Science and Technology, Nirjuli-791109, Arunachal Pradesh, India; madhu.kamle18@gmail.com; 2School of Exercise and Nutrition Sciences, Deakin University, 221 Burwood Hwy, Burwood VIC 3125, Australia; kumar.dipendra2@gmail.com; 3SAB Miller India Ltd., Sonipat, Haryana 131001, India; sheetaldeshwal1993@gmail.com; 4Molecular Genetics Laboratory, Department of Biotechnology, Yeungnam University, 280 Daehak-Ro, Gyeongsan, Gyeongbuk 38541, Korea; keun126@ynu.ac.kr; 5Stemforce, 302 Institute of Industrial Technology, Yeungnam University, Gyeongsan, Gyeongbuk 38541, Korea

**Keywords:** Fumonisins, *Fusarium* spp., food contamination, health issues, secondary metabolites

## Abstract

The fumonisins producing fungi, *Fusarium* spp., are ubiquitous in nature and contaminate several food matrices that pose detrimental health hazards on humans as well as on animals. This has necessitated profound research for the control and management of the toxins to guarantee better health of consumers. This review highlights the chemistry and biosynthesis process of the fumonisins, their occurrence, effect on agriculture and food, along with their associated health issues. In addition, the focus has been put on the detection and management of fumonisins to ensure safe and healthy food. The main focus of the review is to provide insights to the readers regarding their health-associated food consumption and possible outbreaks. Furthermore, the consumers’ knowledge and an attempt will ensure food safety and security and the farmers’ knowledge for healthy agricultural practices, processing, and management, important to reduce the mycotoxin outbreaks due to fumonisins.

## 1. Introduction

Fumonisins are secondary metabolites produced in cereals by pathogenic fungi, namely *Fusarium verticillioides, Fusarium proliferatum*, and related species [1]. Moreover, *Aspergillus nigri* also produces fumonisins in the crop plants of peanut, maize, and grape [2,3,4,5,6]. The maize and maize-based products are most commonly infected with fumonisins besides their presence in several other grains (rice, wheat, barley, maize, rye, oat, and millet) and grain products (tortillas, corn flask, chips) [7,8] which have major influences on health. More than 15 fumonisin homologues have been known and characterized as fumonisin A, B, C, and P [9,10]. Further among fumonisin B, FB1, FB2, and FB3 are most abundant with FB1 being the most toxic form that can co-exists with other forms of fumonisin, i.e., FB2 and FB3 [11]. These (FB1, FB2, and FB3) forms are the main food contaminants. FB1 consists of a diester with propane-1,2,3-tricarboxylic acid (TCA) and 2-amino-12,16-dime thyl-3,5,10,14,15-pentahydroxyleicosane where hydroxyl (OH-) groups at the C-14 and C-15 positions involved with the carboxyl groups (-COOH) of TCA to form an ester. On the other hand, FB2 and FB3 are actually the C-5 and C-10 dehydroxy analogues of FB1 [12]. 

The toxins are linked with several health issues like cancer of the esophagus as evident from different regions of the world. Fumonisins are a very sensitive issue all around the world, which occur in Europe (51%) and Asia (85%) [13]. The occurrence of fumonisins with other related toxins in feed and food is reported in various countries like Argentina [14], Brazil [15], China [16], Italy [17], Portugal [18], Spain [19], Tanzania [20], and Thailand [21]. They are also reported to have toxic effects on the liver and nephron in all the tested animals [22]. In addition, FB1 is implicated with the incidences of hepatocarcinoma, stimulation and suppression of the immune system, defects in the neural-tube, nephrotoxicity, as well as other ailments. It is prominent as a promoter of hepatocarcinoma [23] where its synergistic interactions with aflatoxin B1 (AFB1) has been exhibited in animal models (rainbow trout and rats) for two stages, i.e., initiation and promotion of cancer [24,25,26]. The international agency for Research on Cancer (IARC) characterized FB1 as a group 2B possible carcinogen for human. Besides this, it can cause toxicity in several animals like rats, mice, and rabbits [27]. Further, a temporary maximum tolerable daily intake for fumonisins has been set as 2 μg/kg bw/day based on the lack of any observed adverse effects for nephrotoxicity in male rats by the joint Food and Agriculture Organization (FAO) and World Health Organization (WHO) [28].

## 2. Major Source of Fumonisin

Fumonisins are mainly produced by *F. verticillioides* and *F. proliferatum* and other *Fusarium* spp. The genus *Fusarium,* belonging to the family Nectriaceae, can be found as saprophytes in soil and plants worldwide [29]. *Fusarium* spp. colonize to the rhizospheres of plants and then subsequently enter into the plant system. Furthermore, *F. verticillioides* and *F. proliferatum* are known to be the most common pathogens of maize (*Zea mays*) [30]. Not only the crops, but also many popular ornamental plants (e.g., aster begonia, carnation, chrysanthemum, gladiolus, etc.) are frequently attacked by different *Fusarium* species, viz., *F. oxysporum, F. foetens, F. hostae*, and *F. redolens* at various stages of production [31]. 

*Fusarium*, on the other hand, infects orchids in both pathogenic and non-pathogenic forms. The non-pathogenic forms are either decomposers [32] or in mutual relation where they help in the germination of seeds and the color development of seedlings [33]. The non-pathogenic forms also help to mitigate the infection of *Fusarium* wilt on various crops [34]. Soils responsible for suppressing *Fusarium* wilt are found to be dominant in the *Fusarium* spp. like *F. oxysporum* and *F. solani* which are of agricultural importance [35,36]. The *Fusarium* species infect maize and produce fumonisins mainly at the pre-harvesting stage. Furthermore, fumonisin production has been observed during the post-harvest period; however, under adverse conditions of storage [37]. Dietary exposure of fumonisins can lead to several harmful outcomes in both farm and experimental laboratory animals. For example, these toxins are responsible for leukoencephalomalacia in horses [38], pulmonary edema syndrome in pigs [39], hepatotoxicity and nephrotoxicity in rats [40], and apoptosis in many other types of cells [41].

## 3. Chemistry and Biosynthesis of Fumonisin

Fumonisins (FBs) consist of two methyls (–CH_3_), one amine (-NH_2_), one to four hydroxyl (-OH^-^), and two tricarboxylic ester groups located at different positions along with the linear polyketide-derived backbone. The biosynthesis step comprises the addition of two molecules of tricarballylic esters and one alanine-derived amine to a C-18 polyketide backbone [42]. FBs structural identity has been established, which are similar to sphingosine and are an integral part of cell signaling, growth, and communication [43]. It was believed that fumonisin formation could be controlled by disrupting the biosynthesis of sphingolipids [44]. The biosynthesis process of the toxin has been initiated to illustrate these cellular mechanisms and to design modified analogs [45,46]; however, to date, single total synthesis has been achieved by Pereira et al. [47]. There are intra-specific differences in the biosynthesis of fumonosins depending on the environmental conditions, e.g., temperature, the wavelength of light, humidity, and media composition for both the *Fusarium* spp. *F. verticillioides* and *F. proliferatum* [48]. Even the responses of strains were found to be different when the plant extracts were added from common hosts of *F. proliferatum* [49]. 

## 4. Genes Responsible for Fumonisin Production

Exploring the biosynthesis of trichothecene and fumonisin has revealed the gene cluster fumonisin biosynthetic gene (FUM in *Fusarium* and *Aspergillus*) which is responsible for the production of fumonisins, two transport proteins, and a transcription factor [50]. The expression of these genes is co-regulated and related to the FUM genes expression as well; however, it is influenced by ecological conditions [51,52]. The production of fumonisin is dependent on FUM1 which further expresses an enzyme complex known as polyketide synthase that catalyzes the initial step for fumonisin biosynthesis [53]. Furthermore, a positive correlation has been identified between the proportion of FUM1 transcripts being estimated by real-time RT-PCR and the proportion of fumonisins biosynthesized by the *F. verticillioides* and *F. proliferatum* species [54]. FUM19 lies at a distance of 35 kb downstream of the FUM1 gene that expresses an ATP-binding cassette responsible for exporting extracellular fumonisins [51]. Further, the expression of an aminotransferase by FUM8 functions to maintain the biologically active and mature FB1 molecule [55].

*A. niger* genome has a *Fusarium* FUM cluster homologue consisting of eleven homologues of the *Fusarium* genes namely fum1 (polyketide synthase), fum3, fum6, and fum15 (hydroxylase), fum7 (dehydrogenase), fum8 (aminotransferase), fum10 (acyl-CoA synthase), fum13 (carbonyl reductase), fum14 (condensation-domain protein), fum19 (ABC transporter), and fum21 (transcription factor) genes [56,57]. The FUM cluster in the *A. niger*, also known to have a dehydrogenase gene (sdr1), which is of a short-chain length, is absent in the *Fusarium* FUM cluster and its role in the process of fumonisin biosynthesis is unknown [56,57]. Further, the *Fusarium* FUM2 gene is also absent from the *A. niger* FUM cluster which causes hydroxylation at the C-10 backbone position of fumonisin [58]. The absence of a FUM2 homologue in the *A. niger* cluster has been seen to be consistent with other studies as well revealing that *A. niger* produces fumonisins (FB2, FB4, and FB6) only when it lacks a hydroxyl at C-10 [59,60,61]. In addition to these, genes like FUG1 and FST1 have been also confirmed to have an important role in fumonisin biosynthesis in *F. verticillioides* besides their role in maize kernel colonization [62,63]. Furthermore, Niehaus et al. [64] have identified 21 polyketide synthase (PKS) in the genome of the *F. proliferatum* where PKS3 and PKS11 are predicted to be linked with the biosynthesis of fumonisin.

## 5. Occurrence in Food

The contamination of foods by fumonisin depends on agroclimatic conditions (Table 1). The most commonly infected groups in food are the cereals (rice, wheat, barley, maize, rye, oat, and millet). The FB1 has been reported to contaminate numerous food products like asparagus and garlic [65], barley foods [66], beers [67], dried figs [68], and milk [69]. Maize (*Zea mays* L.) and maize-based products are one of the most commonly infected foods by FB1 [70]. Maize is used for manufacturing several products like tortillas and tortilla chips, corn flakes and corn starch, popcorn, grits, flour, and oils. However, the contamination by FB1 and FB2 is decreased by 59% during the manufacturing of tortilla chips from maize flour, while 60% for flour and 50% for grits and snack products due to the heat treatment by extrusion [71]. Further, several other products like cornflakes [72], the Portuguese maize bread [73], tea (black and herbal), along with some medicinal plants [74] have also been contaminated by fumonisins.

## 6. Effects on Agriculture and Food

Annually 25% of harvested crops are contaminated by mycotoxins, causing huge economic losses to agricultural and industrial commodities. These mycotoxins are stable in nature and do not eliminate during food processing, cooking, baking, roasting, and pasteurization. The meagre agricultural, as well as post-harvest practices like inappropriate drying techniques, handling procedure, packaging materials and methods, and storage and transport conditions, are responsible for the increased risk of fungal growth and fumonisin contamination [106]. Cao et al. [107] investigated the accumulation of fumonisins at different kernel developmental stages as well as during the drying of the kernel of hybrid varieties of white maize. They observed *Fusarium* (especially *F. verticillioides*) to be the most prevalent genus for growth and contamination as compared to *Aspergillus* and *Penicillium*. The lower humidity of kernels favoured damage by insects along with fungal growth and accumulation of fumonisins [107]. The occurrence of fumonisins have been reported in edible plants like onion, garlic, asparagus, and pea seed [108,109]; in other cereals, mainly in wheat [84,110] as well as in crops like sorghum, beans (white, adzuki, mung), barley, soybean, asparagus spears, and figs [111,112]. Besides this, fumonisins have been found to impact the performance of aquatic animals like the Nile tilapia fingerlings and juveniles [113]. Fumonisins affected the hepatic expression of growth hormone receptor (GHR) and insulin like growth factor 1 (IGF-1) in these species, which is an indication that other aquatic animals and plants could also be affected by fumonisins posing a serious threat to food safety and security.

Fumonisins are an important class of mycotoxins produced by *F. proliferatum* and *F. verticillioides* along with others such as *F. napiforme, F. oxysporum, F. dlamini, F. nygamai* and *F. anthophilum* that are widely distributed, having potential health hazards to humans and animals [9]. These toxins are widely distributed in crops like corn, rice, sorghum, barley, and coffee. The exact causes of ear rot and kernel rot diseases is not well known but may be due to changes in weather such as dry weather followed by warm wet weather during flowering. The damage caused by the insect at the time of maturity allows strains present in nature to enter the ear and kernels. Rain before harvest may intensify the contamination of fumonisins in corn. Sometimes there are substantial amounts of fumonisins present in the non-symptomatic kernels of corn [114]. Yoshizawa et al. [103] reported the occurrence of fumonisins and aflatoxins in eighteen samples of corn in Thailand and found FB1 and FB2 and isolated *F. moniliforme* and *F. proliferatum* from the corn grit samples. Studies carried out in the USA reported the presence of FB1 and moniliformin in 34% of corn samples and 53% of corn-based food products, respectively [115]. A study in Brazil was conducted (during 2007–2010) to detect fumonisins in corn-based food products and reported that FB1 and FB2 were present in 82% and 51% of the examined products, respectively [116]. Contaminations by FB1 and FB2 observed in poultry broiler and feed fatting calves in South Korea [117]. Abdallah et al. [118] found the co-occurrence of FB2 and ochratoxin A and B in the date palm. In Brazil, it was reported that the production of fumonisins by *F. verticillioides* is found in both symptomatic and asymptomatic grains [119]. 

Furthermore, a survey was conducted in Japan for aflatoxin, ochratoxin A, and fumonisins contamination using HPLC and LC-MS. Results revealed that peanut butter is contaminated by aflatoxin, while orchratoxin A infection in oatmeal, rye, buckwheat flour, green coffee beans, roasted coffee beans beers, wheat flour, and wine. However, fumonisins were observed in popcorn, frozen corn, corn flasks, and corn grits [120]. Noonim et al. [59] analyzed the aflatoxin and fumonisin contaminations in different samples of Thai dried coffee, and it was noted that no *Fusarium* spp. were observed; however, *A. niger* was present in the coffee beans and produced fumonisins along with aflatoxins. A variable range of acetyldeoxynivalenol, deoxynivalenol, neosolaniol, fumonisin B1, and ochratoxin A contaminations were observed in Spanish coffee, and this variation was due to different methods of coffee brewing [121].

## 7. Mechanism of Toxicity and Health Effects of Fumonisins

### 7.1. Mechanism of Toxicity

FB1 predominates in 70% of the total FBs naturally occurring in infected food and feed samples [122]. FB1 express both acute and chronic symptoms in infected animals. FB1, though being an initiator of cancer, is non-genotoxic [123]. The major organs affected are liver and kidney; however, the severity of infection depends upon the strain and species [124]. The intestine, on the other hand, is a possible target for fumonisin toxicity [125]. FBs contamination has raised higher concern because of their interference with sphingolipid metabolism that ultimately leads to serious health concerns. Fumonisins are also linked to esophagal cancer and defects of the neural tube in humans [126]. Further, FB1 is the major causative agent for porcine pulmonary edema (PPE) [39], the toxicity of the liver and nephron in rodents [127], as well as cancers of the liver and esophagus in humans [128]. 

Franceschi et al. [129] studied the relationship between maize consumption and the risk of cancer of the upper digestive tract in the Pordenone Province in the north-eastern part of Italy. The population of this province has a high incidence of these neoplasms and shows particularly elevated levels of alcohol and tobacco use, in addition to high maize consumption. They observed that there were highly significant associations with frequent intake of maize emerging for oral cancer, pharyngeal cancer, and esophageal cancer. Dragan et al. [130] showed that the FB1 caused renal carcinomas in male rats and liver cancer in female mice. FB1 also induces apoptosis in many kidney cell lines, primary cell cultures, and also in vivo in rat liver and kidney [130,131]. Sun et al. [132] reported high contamination of FB1 in the food of the Huaian and Fusui city of China and suggested that FB1 may have a contributing role in human esophageal- and hepatocarcinogenesis. Further, Alizadeh et al. [133] studied 66 samples of both corn and rice from the Golestan province of Iran and observed high levels of FB1 contamination in both corn (223.66 µg/g) and rice (21.59 µg/g). They found a significant relationship between FB1 contamination in rice and the risk of esophageal cancer. Besides this, FB1 was found to be toxic to other cell lines. For example, FB1 triggers dose-dependent apoptosis and necrosis in esophageal carcinoma (SNO) cell lines in humans. Similarly, FB1 inhibited the activity of ceramide (CER) synthase, which is responsible for the acylation of sphinganine (Sa) and the recycling of sphingosine (So). This leads to an increment in the intracellular cytotoxic Sa-compound. Therefore, the variation of Sa/So proportions in urine and blood samples may denote the exposure of FBs in several animals; however, this has not been accurately validated [134].

### 7.2. Health Effects of Fumonisin

Equine leucoencephalomalacia first reported in 1891 is now revealed to be caused by consuming fumonisin-contaminated maize [135]. Further, the consumption of maize culture material infected by *F. verticillioides* [136] is responsible for the occurrence of porcine pulmonary edema (PPE) [30]. Since then, the outbreaks of PPE in the USA have been identified because of fumonisin infection. Further intake of fumonisin-affected diets by pregnant women causes neural tube defects in the developing fetus [126,137]. Sadler et al. [138] reported that FB1 has the potential to inhibit embryonic sphingolipid synthesis, produce embryotoxicity, and block folate transport and has been associated with increased prevalence of cancer and neural tube defects. On the other hand, Missmer et al. [126] reported the prevalence of neural tube defects (NTDs) doubled between 1990–1991 in Mexican–American women because they consume large amounts of corn in the form of tortillas, due to which they may be exposed to high levels of fumonisin. Fumonisin exposure increases the risk of NTDs and a dose above the threshold level may cause fetal death. Similarly, the exposure of fumonisin and its effect on esophageal and liver cancer is rare [132,139]. While no direct evidence of fumonisin hazard is found, its prolonged exposure may lead to cancer and birth defects in humans [140]. Moreover, the co-contamination of foods by fumonisin and aflatoxin has imposed risks of occurrences of outbreaks in southwest Nigeria [140], and the rural areas of Malawi in sub-Saharan Africa [141].

Besides this, the contamination of breast milk by fumonisins has been reported in several studies [142,143,144]. Recent studies have revealed the relationship between exposure to FBs and growth impairment in children [145,146,147]. According to Shirima et al. [146], fumonisin exposure negatively impacted child growth among children in Tanzania, which was confirmed based on urinary biomarker levels of fumonisin (UFB1). On the other hand, aflatoxin exposure had no significant impact on child growth. Furthermore, breastfeeding and weaning practices were considered to be associated with growth impairment in children due to exposure to FB1 [147]. The fumonisin carry-over has been observed in cow’s milk as well [69]. Therefore, the incidence of fumonisin in human breast milk and its consumption by infants cannot be ignored, as the milk is a crucial part of infants’ nutrition [148]. 

## 8. Effects of Processing on Fumonisin

Fumonisins are known to be comparatively heat-stable and affected only when heated above 150–200 °C during food processing techniques like baking, frying, roasting, or extrusion cooking. The degree of reduction in their chemical structure and toxicity depends on the cooking conditions and the composition of the food matrix [149]. However, this reduction could be due to the structural modifications of fumonisins while interacting with other components of food that leads to the conjugate’s formation [150]. FB1 interacts with reducing sugars to form strong covalent bonds during heat treatments. For instance, FB1 reacts with D-glucose of corn grits during the extrusion cooking at 160–180 °C and forms a reaction product, N-(carboxymethyl) fumonisin B1 (NCM) [151]. However, the condensation reaction of FB1 and D-glucose forms N-(deoxy-Dfructos-1-yl) FB1 (NDF) [152]. 

Besides this, the wet milling causes the reduction of fumonisins to some extent in steep water. Further industrial milling processes reduce the fumonisin content significantly such that the fractions obtained (gluten, fiber, germ, and starch) are suitable for animal and human consumption [153]. However, during the dry milling process, there is a negligible reduction in fumonisin content as the fumonisins are embedded in the germ and pericarp in higher concentrations than in the endosperm and its derivatives [72,154,155]. Fumonisins are variably distributed in cereals and the fractions depending upon the type of cultivars, agricultural practices, and the method of milling processes [153,156]. The toxins might be degraded or modified during the processing of Tortillas at high temperatures and pH [157]. However, the industrial processing methods like roasting, frying, and extrusion cooking are effective in reducing the fumonisins to significantly low levels [158].

## 9. Effects of Environmental Temperature on Fumonisin Production

The two main factors impacting on the growth of fungus and the production of fumonisin are temperatures and water potential [159]. Therefore, the toxins are predominant in temperate and Mediterranean climatic regions [160,161,162,163]. The Mediterranean climate regions experience extreme temperature, rainfall patterns, as well as longer durations of drought. These conditions might lead to variation in the population of mycotoxigenic fungi and the fumonisin production by them which ultimately impacts the control strategies [164]. The infection of maize by *F. verticillioides* and accumulation of fumonisins is determined by the climatic conditions, insect damage, as well as the plant characteristics. The ear rot infection by *F. verticillioides* occurs during the flowering stage and is favored by warm and dry conditions; however, both warm and wet conditions following silking have been found to be favorable for disease development [165]. The weather conditions are critical for toxin accumulation during flowering as well as prior to harvesting [166,167]. It has been found that the less rainfall with maximum temperatures of 30–35°C during flowering induces disease development [168].

Cendoya et al. [169] evaluated the effect of different levels of temperature and water activity (a_w_) on the fungal growth and fumonisin biosynthesis in wheat using three strains of *F. proliferatum.* Temperatures of 15, 25, and 30 °C and a_w_ of 0.99, 0.98, 0.96, 0.94, 0.92, and 0.88 were evaluated. They found maximum growth of fumonisins at the highest a_w_ of 0.99 at 15 °C for two strains while for the third strain, the maximum growth was observed at 25 °C at the same a_w_ level. Furthermore, environmental factors like light along with nutrients available impacted the growth of *F. proliferatum* and the production of fumonisin [48,170]. In addition, Li et al. [171] evaluated the impact of pH levels on the growth of *F. proliferatum* culture. It was found that the toxin production was significantly inhibited in culture maintained at pH 5 compared to the culture at pH 10. However, the acidic pH 3–4, was found to enhance FB1 production by the fungus *F. proliferatum* [172].

## 10. Detection Techniques

The FB1 presence was detected by the Association of Official Analytical Chemists (AOAC) official method in food and feed samples. The derivatization was done using precolumn with ortho-phthaldialdehyde (OPA) and the detection by chromatographic techniques like HPLC (high-performance liquid chromatography) coupled with a fluorescence detector (HPLC-FLD). However, the drawback of this method is the use of high sample size (around 50 g), more extraction solvent (methanol:water), and solid-phase extraction (SPE) cartridges [173]. Therefore, methods like QuEChERS (Quick, Easy, Cheap, Effective, Rugged, and Safe) proved to be ideal for the detection of FB1 [174,175,176]. Some of the commonly used techniques for fumonisin extractions include: (i) solid-liquid extraction (SLE) [177,178,179,180], (ii) liquid-liquid extraction (LLE) [181,182], (iii) matrix solid-phase dispersion (MSPD) [87,183,184], and (iv) dispersive liquid-liquid microextraction (DLLME) [185]. Recently, it was observed that the extraction yields were higher in finer flours indicating the importance of sample particle size on the recovery of fumonisins [11]. 

The traditional analytical methods to detect and quantify fumonisin include HPLC or UPLC (ultra-performance liquid chromatography) coupled with detectors such as UV–Vis spectrophotometric [186], fluorescence [187,188], and mass spectrometry (MS) [176,189,190,191]; liquid chromatography-mass spectrometry (LC-MS), and thin-layer chromatography (TLC) [192,193,194]. As these methods are expensive, tedious, and time-consuming [195], other advanced methods like the detection of mycotoxins producing fungi, enzyme-linked immunosorbent assay (ELISA), surface plasmon resonance (SPR), lateral flow immunoassay (LFI), immunosensors, electronic nose, and hyperspectral imaging are found to be more efficient [194,196]. Fumonisins producing genes have been amplified by PCR to detect *Fusarium* species in freshly harvested maize kernels [197]. PCR-based methods are used for the detection of mycotoxins producing fungal genera *Fusarium, Aspergillus*, and, *Penicillium* [198,199].

Recently, Nagaraj et al. [196] used a multiplex PCR technique to detect fumonisin producing *F. verticillioides* strains. ELISA coupled with PCR, i.e., PCR-ELISA by Omor et al. [200] for the detection of *F. verticillioides* based on the FUM21 gene in corn. In addition to this, a highly sensitive indirect competitive enzyme-linked immunosorbent assay (icELISA) and gold nanoparticle-based gray imaging quantification immunoassay (GNPs-GI) has been developed to detect FB1 in agricultural products [201]. Another important and non-destructive way of identifying toxigenic fungi in maize is by the application of hyperspectral imaging processes [202,203]. Besides this, the color-encoded lateral flow immunoassay (LFIA) has emerged as a leading technique for simultaneous detection of aflatoxin B1 and type-B fumonisins in a single test line [204]. Nowadays, electrochemical immunosensors are employed for rapid and sensitive detection of FB1 [205]. Furthermore, a rapid and ultrasensitive molecularly imprinted photoelectrochemical (MIP-PEC) sensing technique has been recently developed to measure FB1 [206].

## 11. Masked Mycotoxins as a major concern in detection

The masked mycotoxins issue was initially seen during the mid-1980s due to several mysterious cases of mycotoxicosis occurrence; however, the symptoms of mycotoxins in affected animals did not connect with the low mycotoxins content detected in their feed. At the same time, the metabolic biotransformation of deoxynivalenol (DON) to the less toxic derivatives *in planta* was first reported to appear in corn inoculated with *F. graminerium* [207] and also in naturally infected winter wheat [208]. In vivo studies for masked mycotoxins were carried out in pig and reported that zearalenone-14-glucoside was decomposed during the digestion process and zearalenone (ZEN) and zearalenol (ZEL) were detected in urinary and fecal metabolites [209].

During infection in plants, the mycotoxins produced by fungi are modified by plant enzymes and often conjugated to more polar substances, like sugars. These form of toxins are often less toxic metabolites stored in the vacuole in the soluble form or bound to macromolecules and are not detectable during routine analysis processes; therefore, referred to as masked mycotoxins [210]. These mycotoxins may not be a homogeneous group of contaminants but somewhat a complex mixture of different plant metabolites of various classes of mycotoxins and they are overall termed as the ‘maskedome’ [211]. Detection of masked mycotoxins is difficult as they change the physiological properties of their molecules leading to modified chromatographic behavior [212]. Due to less detectability, these toxins are a serious concern for food safety and these toxins may be converted back to the parent toxin forms during the food digestion process [213]. De Boevre et al. [214] analyzed cereal-based food products and raw feed materials for the presence of mycotoxins including deoxynivalenol, 3-acetyldeoxynivalenol, 15-acetyldeoxynivalenol, zearalenone, α-zearalenol (ɑ-ZEL), β-zearalenol, and their respective masked forms like α-zearalenol-1-3glucoside, zearalenone-4-glucoside, α-zearalenone-4-glucoside, β-zearalenone-4-glucoside, and zearalenone-4-sulfate in fiber-enriched bread, bran-enriched bread, cornflakes, popcorn, and oatmeal. Binder et al. [215] evaluated the absorption, distribution, metabolism, and excretion (ADME) of plant (ZEN-14-Glc, ZEN-16-Glc) and fungal (ZEN-14-S) ZEN metabolites in pigs and found that the total amounts of ZEN-14-GlcA, ZEN, and α-ZEL were excreted into urine after 0–48 hours of administration.

## 12. Degradation Kinetics

The degradation of FB1 was first revealed by Duvick et al. [216] to occur by microbes like *Exophiala spinifera, Rhinocladiella atrovirens,* and *Sphingomonas* or *Xanthomonas* having the capacity to metabolize FB1. These microbes were isolated from various tissues of maize. Further, the fumonisin metabolism by *E. spinifera* and the bacterium (deposited as ATCC55552 with the American Type Culture Collection) was studied by radiochemical and chromatographic (e.g., thin layer chromatography, TLC) methods. The initial two steps of biodegradation of FB1 were revealed to be due to de-esterification by a carboxylesterase releasing two tricarballylic acid (TCA) moieties leading to the formation of hydrolyzed FB1 (HFB1). The bacterial strain ATCC55552 further metabolized 14C-FB1 with the release of 14 molecules of CO_2_. However, *E. spinifera* could not further metabolize the TCA moieties. Blackwell et al. [217] later studied the oxidative deamination process of HFB1 by *E. spinifera* through TLC and mass spectrometry. They found that the HFB1 gets converted to Nacetyl HFB1 and 2-oxo-12,16-dimethyl-3,5,10,14,15-icosanepentol hemiketal. A cluster of genes in the bacterium ATCC55552 responsible for the degradation of fumonisin is mentioned in a patent, WO 00/04158 by Duvick et al. [218]. 

FB1 can be degraded to the less toxic form of hydrolyzed FB1 (HFB1) by an enzymatic process which could be used to reduce intestinal inflammation in pigs [219]. Further, the gene that catalyzes the oxidative deamination process of HFB1 in *E. spinifera* was revealed; however, the responsible enzyme for the deamination reaction is still unknown [218]. Later, Benedetti et al. [220] screened and isolated a bacterium related to the Delftia/Comamonas group (known as NCB 1492) from the soil. It was able to hydrolyze and deaminate FB1, but still, the sequences of the responsible genes are unknown. A year before, *Sphingomonas* sp. MTA144 was shown to have fumonisin degrading activity [221]. Further, Heinl et al. [222] identified two genes (carboxylesterase and aminotransferase) having prominent fumonisin-degrading activity. In addition to this, essential oils from plants were found to inhibit as well as degrade FB1 for example anise, camphor, cinnamon, citral, clove, eucalyptus, *Litsea cubeba*, and spearmint [223,224].

## 13. Management and Control Strategies

### 13.1. Management and Control using Agricultural Practices

As the crop plants like maize are infected by fumonisins during their growth in fields [225], the implementation of good agricultural practices (GAP), good storage practices (GSP), and good manufacturing practices (GMP) can mitigate the fumonisin contamination [226]. Harvesting the crop at earlier stages could be one of the strategies to control fumonisin contamination [227]; however, this cannot be applied to crops that need to be harvested at full maturity. Instead, the early harvest can be done for forage maize to increase the digestibility of silage. These practices require careful study as the farmers prefer a delayed harvest because of advancement in technologies. For instance, the use of kernel processors during forage harvesting leads to the production of digestible silage from maize when harvested at later stages [228]. 

Recently, the Codex Alimentarius Commission has set maximum levels of fumonisin at 4000 μg/ kg and 2000 μg/kg, respectively for raw maize and for maize flour and meal which have been implemented in South Africa. However, the lowering of fumonisin exposure in subsistence farmers need an integrated approach, and this cannot be solely achieved by regulatory measures [229]. Besides these approaches, the use of nanotechnology and genetic engineering should be encouraged in the field of agriculture to develop resistant varieties of crops to get rid of *Fusarium* infection and FB contamination. The creation of drought and insect-resistant crops can also play a significant role in the fumonisin control as these factors are responsible, in one or the other way, for the fungal infection [230]. In addition to this, educating the farmers about the importance of drying and sorting out of the contaminated kernels from the crops can manage and control the risk of infection to some extent [231]. The in vitro study of combinations of fungicides (fludioxonil + metalaxyl-M) showed that it was not sufficient in the growth inhibition of *F. verticillioides* and even the increase in the production of FB1 by their strains [232]. A similar study also showed that these fungicides inhibit the growth and extracellular material formation but enhance the sporulation and fumonisin production in liquid culture of *F. verticillioides* [233]. Masiello et al. [234] reported that prothioconazole and thiophanate-methyl were effective in reducing the *F. graminearum* (52% and 48%) and *F. proliferatum* contamination (44% and 27%) under the field trial.

Fumonisin production and *Fusarium* growth are the result of interactions with various biotic and abiotic factors. In the case of abiotic factor temperature, water stress was the most significant environmental factor which influenced the fumonisin production and *Fusarium* growth. Several other stress conditions such as osmotic stress, pH, and fungicides were reported for the production mycotoxins [235,236]. *F. verticillioides* isolates were found to exhibit better performance at higher temperatures and under water stress conditions in comparison to *F. proliferatum*, another fumonisin-producing species. Marin et al. [237] suggested that environmental conditions leading to water stress (drought) might result in an increased risk of fumonisin contamination of maize caused by *F. verticillioides*. Drought stress and excess irrigation favor *Fusarium* infection. Drought stress should be avoided during the period of wheat seed development and maturation [238]. Excess moisture during the flowering seasons and early grain-fill periods also supports the *Fusarium* infection and moisture also increases the DON contamination [239]. Fungicide treatments were found to be effective against wheat *Fusarium* infection and DON contaminations [240,241]. Azole fungicides were found to be effective in the reduction of DON and other emerging and modified mycotoxins [242]. Therefore, an integrated approach, involving good agricultural management practices, hazard analysis, and critical control point production, storage management along with selected biologically based treatments, and mild chemical and physical treatments could reduce the fumonisin contamination effectively [243].

### 13.2. Management and Control using Mycotoxin Binder

Mycotoxin binders or adsorbents are substances that bind to mycotoxins and prevent them from being absorbed through the gut and prevent their entrance into the blood circulation. The mycotoxin binders can be helpful and utilized when other preventive measures fail against molds and mycotoxins [244]. The main aim of mycotoxin binders is to prevent the absorption of the mycotoxins from the intestinal tract of animals by absorbing the toxin to their surface. These binders may be organic or inorganic in nature, such as clay and yeast derived products, respectively [245]. However, mycotoxin modifiers are used to alter the chemical structure of mycotoxins and reduce their toxicity. These are microbiological in origin containing whole bacterial and yeast culture and specifically extracted compound such as enzymes [246]. In the field during harvesting of the crop, the production of mycotoxins can be reduced by choosing varieties that are adapted to the growing area and have resistance to fungal diseases. Mycotoxin production can also be reduced in the field by proper irrigation and balanced fertilizer applications [247]. These binders bind to the mycotoxins strong enough to prevent toxic interactions with the consuming animals and their absorption across the digestive tract. Potential absorbent materials include activated carbon, aluminosilicates (bentonite, clay, montmorillonite, zeolite, pollyosilicates etc.), complex indigestible carbohydrates (Cellulose, polysaccharides in the cell wall of yeast and bacteria such as glucomannans, petidoglycans) and other synthetic polymers such as cholestryamine and polyvinylpyrrolidone and derivatives [247]. De Mil et al. [248] characterized 27 feed additives marketed as mycotoxin binders and screened them for their in vitro zearalenone (ZEN) adsorption. Recent studies showed that the addition of the commercial toxin binders to the aflatoxin B1 (AFB1) containing diets reduced the adverse effects of AFB1 and could be helpful as a solution to the aflatoxicosis problem in young broiler chicks [249].

## 14. Conclusion

The contamination of food and feed by fumonisin is a serious threat for disease outbreaks worldwide. The various techniques ranging from physical to biochemical as well as genetic engineering can be utilized in an efficient manner to mitigate fumonisin contamination of foods. However, a major issue of concern lies with the development of fungal and insect resistant crops to combat the fungal infection and fumonisin contamination. The naturally occurring soil microorganisms have been reported to have an immense capability of degrading and reducing the biosynthesis of fumonisins and its contamination in various agricultural crops. Moreover, the application of nanotechnology and genetic engineering should be given more emphasis to develop resistant varieties of crops and ensure the safety and quality of food for future generations.

## Figures and Tables

**Table 1 toxins-11-00328-t001:** Occurrence of Fumonisin B1 and FB2 in cereals and cereal-based foods around the world.

Country	Food Matrix	FB1 (Range, µg/kg)	FB2 (Range, µg/kg)	Detection Technique	Reference
UK	Corn	200–6000	-	TLC	[75]
The Netherlands	Corn flour	40–90	-	HPLC	[76]
Switzerland	Corn grits	0–790	0–160	HPLC	[77]
Turkey	Cornmeal	250–2660	550	HPLC	[78]
Ghana	Corn	11–1655	10–770	HPLC	[79]
Malawi	Corn	20–115	30	HPLC	[80]
Zambia	Corn inbred lines	20–1420	10–290	HPLC	[81]
Bahrain	Corn kernel	25	-	HPLC	[82]
Kenya	Corn kernel	110–120	-	HPLC	[83]
Venezuela	Yellow corn	40–15,050	-	HPLC	[84]
Korea	Corn for popping	23–1210	-	direct competitive (dcELISA) and HPLC	[85]
India	Corn seed samples	133 to 1617	-	HPLC	[86]
Iran	Corn	10–3980	<10–1180	HPLC	[87]
Thailand	Corn	63–18,800	50–1400	HPLC	[88]
Nepal	Corn kernels	50–4600	100–5500	HPLC	[89]
Indonesia	Corn kernels	51–2440	<376	HPLC and GCMS	[90]
Argentina	Durum wheat	10.50–987.20	15–258.50	HPLC-MS/MS	[91]
Brazil	Wheat	958–4906	-	HPLC-FL	[92]
Canada	Wheat	-	-	HPLC	[93]
Central Europe	Wheat/wheat bran	-	-	ELISA	[94]
China	Wheat flour	0.30–34.60	-	UPLC-MS-MS	[95]
France	Organic Oat, rye and wheat flakes with maple syrup	75.70–98.10	62.10–81.10	HPLC-MS/MS	[96]
Germany	Organic wheat flakes	20.20–59.80	25.40–41.80	HPLC-MS/MS	[96]
Iran	Stored wheat samples	15–155	12–86	HPLC	[97]
Italy	Cereals, whole meal flours	10–2870	10–420	LC-MS	[98]
Japan	Wheat	>10	-	LC-ESI-MSMS	[99]
Serbia	Wheat	750–5400	-	ELISA	[100]
South Africa	Wheat and wheat products	1000–30,000	-	TLC, HPLC, Ms/MS	[101]
South America	Wheat/wheat bran	-	-	ELISA-HPLC	[94]
South-East Asia	Wheat/wheat bran	-	-	ELISA-HPLC	[94]
Southern Europe	Wheat/wheat bran	-	-	ELISA-HPLC	[94]
Spain	Wheat Gofio	787.50–1001.40	645.20–952.10	HPLC-MS/MS	[96]
Syria	Durum wheat	5–6	12	HPLC-MS/MS	[102]
Tunisia	Wheat-based products	88.33–184	121–158	LC-MS/MS	[103]
United States	Wheat	5–2210	2-249	LC-MS	[104]
Zimbabwe	Wheat	2500–6000	-	HPLC	[105]

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
