# Peer review of "Fumonisins: Impact on Agriculture, Food, and Human Health and their Management Strategies"

_toxins, 2019, doi:10.3390/toxins11060328_

Round 1
Reviewer 1 Report
I found a short but very nice and informative review. The introduction can be a bit more expanded. In the text and in the reference list the name of some microorganisms (especially Fusarium) should be italicized.
Author Response
We are really thankful to the reviewer for valuable suggestions.
As per instructions all the necessary changes are made and highlighted blue in color.
The text has been added in Introduction highlighted in blue. All the text and references list has been corrected for Fusarium and other species as suggested.
Reviewer 2 Report
Mycotoxins are a very interesting topic. So, a review about the research in this field certainly has an added value. However, there are a lot of typos in this manuscript and this review only considers a few topics. Furthermore, it does not have an added value compared to the existing reviews on FBs. However, I suggested resubmission after major revision, since I really believe in the suggested title and topic. I indicated some suggestions in the pdf. Below you can find some things which should be added.
Add more on the detection methods only HPLC is described
Add something about agricultural practices e.g. fungicides, but also fertilizers, etc.
What about masked mycotoxins?
What about mycotoxin binders?
What about the effect of the presence of e.g. Aspergillus on the toxin production of F. verticillioides
And in the rebuttal I want to see what is new compared to existing reviews, indicated in which fields your has an added value

Author Response
Response to Reviewer 2:
o We are really thankful to the esteemed reviewer for the valuable comments and suggestions which are helpful in the improvement of the quality of the current manuscript. We incorporated all the suggested corrections and literature as per instruction. All necessary changes and incorporation of matters highlighted blue in color.
o Some more recent detection methods besides HPLC added in the Detection section highlighted in blue.
o Agricultural practices added in section 6. Effects on Agriculture and Food
o Masked mycotoxins incorporated in a separate section 11.
o Mycotoxin binders added in section 13.2
o Effects of the presence of Aspergillus on toxin production mentioned at the end of 6.
o Effects on Agriculture and Food highlighted in blue.
Comments: Also in the American countries FBs are problem + add references for each country
Response: The list of countries added in the second paragraph of the introduction marked in blue.
Comments: In case you divide your section (3.) into subsections 3.1., 3.1 etc. you should start with the first subsection directly under the main section 3. (which is not the case here).
Response: The corrections can be seen in section 2 of the revised file since the section 3 has been merged with section 7. Mechanism of Toxicity and Health Effects of Fumonisin
Comments: In case you divide your section (3.) into subsections 3.1., 3.1 etc. you should start with the first subsection directly under the main section 3. (which is not the case here).
Response: The corrections can be seen in section 2 of the revised file since the section 3 has been merged with section 7. Mechanism of Toxicity and Health Effects of Fumonisin
Comments: In my opinion F. graminearum does not produce fumonisins so not necessary to mention here.
Response: The sentence has been deleted in the revised manuscript.
Comments: In my opinion these Fusarium species are also considered as pathogens see e.ghttps://onlinelibrary.wiley.com/doi/full/10.1111/mpp.12289
Response: The changes has been incorporated and marked in blue in section 2.
Comments: In section- Outbreaks due to Fumonisin you also write a lot about the effects of toxins, I suggest to merge these sections to avoid repetition.
Response: Outbreaks due to Fumonisin has been merged with section 7. Mechanism of Toxicity and Health Effects of Fumonisin
Comments: In L you write Currently, there are more than 15 homologues of fumonisin .... What is correct?
Response: more than 15 homologues of fumonisin is correct which is mentioned in the introduction part and corrected elsewhere accordingly
Comments: Strain, species, sex... this is a strange summary, since the first two items belong to the fungus and the next to the person/animal?
Response: “strain, species, and sex-dependent differences in dose” correction included in section 7.
Comments: what are high temperature, please specify a range for ear rot infection
Response: The changes has been incorporated and highlighted in blue in section 9.
Comments: Grammatical and other errors marked in the pdf file
Response: All the comments have been been addressed which can be seen in the blue text added in the revised manuscript.
Round 2
Reviewer 2 Report
Dear
There are still plenty of typos in this manuscript, I indicated some of them in the attached pdf. In addition, there is an inconsequent use of terms and sometimes there are even mistakes in the manuscript e.g. ELISA, which is a technique based on DNA dectection, ... and that you can use PCR to quantify toxins... So again, although the topic is interesting, a manuscript with so many typos illustrates that the scientific level is also not very high and you mainly focus on the typos and not on the real content, so again I suggest a major revision!

Author Response
There are still plenty of typos in this manuscript, I indicated some of them in the attached pdf. In addition, there is an inconsequent use of terms and sometimes there are even mistakes in the manuscript e.g. ELISA, which is a technique based on DNA dectection, ... and that you can use PCR to quantify toxins... So again, although the topic is interesting, a manuscript with so many typos illustrates that the scientific level is also not very high and you mainly focus on the typos and not on the real content, so again I suggest a major revision!
Response: Thank you very much for your scientific concern and help in removing the errors. We incorporated all the suggested corrections and grammatical errors marked in PDF file.
Comments: in line 21 you say that maize based products are the most important products contaminated with toxins, but here it seems that all cereals have the same chance of being infected.
Response: The maize and maize-based products are most commonly infected with fumonisins besides their infection in several other grain and grain products.
Comments: How can you explain that growth decreases with a decrease in aw, while above it is mentioned that dry weather stimulates growth?
Response: The optimal aW and temperature range for growth of Fusarium that permitting sub-optimal growth, because Fusarium spp. present on a substrate for longer duration during which optimal aW may change, or temperature fluctuations may occur. While, during dry conditions, changes in relative humidity, temperature and rainfall together stimulates the colonization of F. proliferatum in developing grains.
Comments: How can PCR be used to detect mycotoxins.
Response: Sorry for the incorrectly written lines. The PCR based techniques used for the detections and identification of fungal genera producing mycotoxins. The changes have been made in the corrected copy.
Comments: ELISA is not a technique to detect DNA there are different ELISAs.
Response: Thanks for the correction.
The PCR-ELISA technique used for the detection of FUM21 gene for F. verticillioides. [Omori, A. M.; Ono, E. Y. S.; Bordini, J. G.; Hirozawa, M. T.; Fungaro, M. H. P.; Ono, M. A. Detection of Fusarium verticillioides by PCR-ELISA based on FUM21 gene. Food Microbiol. 2018, 73, 160-167 ]
Round 3
Reviewer 2 Report
The manuscript has been improved compared to the first version, however I still detect a lot of typos (see remarks in the pdf) together with some other suggestions.

Author Response
Thank you very much for your scientific concern and we are really thankful to the esteemed reviewer for the valuable suggestion and corrections. All the necessary changes have been made and the text has been revised as per your kind suggestion.
We pay our sincere thanks to the reviewer for his/her pain for the betterment of the manuscript.